



# OpenMRG: Open data from Microwave links, Radar, and Gauges for rainfall quantification in Gothenburg, Sweden

Jafet C.M. Andersson [1], Jonas Olsson [1], Remco (C.Z.) van de Beek [1], Jonas Hansryd[2]

[1] Swedish Meteorological and Hydrological Institute (SMHI), 601 76 Norrköping, Sweden
[2]Ericsson AB, Lindholmspiren 11, 412 56 Göteborg, Sweden

*Correspondence to*: Jafet Andersson (jafet.andersson@smhi.se)

## Abstract

Accurate rainfall monitoring is critical for sustainable societies, and yet challenging in many ways. Opportunistic monitoring
using commercial microwave links (CML) in telecommunication networks is emerging as a powerful complement to
conventional gauges and weather radar. However, CML data are often inaccessible or incomplete, which limits research and
application. Here, we aim to reduce this barrier by openly sharing data at 10-second resolution with true coordinates from a
pilot study involving 364 bi-directional CMLs in Gothenburg, Sweden. To enable further comparative analyses, we also
share high-resolution data from 11 precipitation gauges and the Swedish operational weather radar composite in the area.
The article presents an overview of the data, including collection approach, descriptive statistics, and a case study of a high-
intensity event. The results show that the data collection was very successful, providing near-complete time series for the
CMLs (99.99%), gauges (100%) and radar (99.6%) in the study period (June–August 2015). The bandwidth consumed
during CML data collection was small, and hence the telecommunication traffic was not significantly affected by the
collection. The gauge records indicate that total rainfall was approximately 260 mm in the study period, with rainfall
occurring in 6% of each 15-minute interval. One of the most intense events was observed on 28 July 2015, during which the
Torslanda gauge recorded a peak of 1.1 mm min$^{-1}$. The variability of the CML data generally followed the gauge dynamics
very well. Here we illustrate this for 28 July, where a nearby CML recorded a drop in received signal strength of about 27
dB at the time of the peak. The radar data showed a good distribution of reflectivities for mostly stratiform precipitation, but
also contained some values above 40 dBZ, which is commonly seen as an approximate threshold for convective
precipitation. Clutter was also found and was mostly prevalent around low reflectivities of −15 dBZ. The data are accessible
at https://doi.org/10.5281/zenodo.6673751 (Andersson et al., 2022). We believe this Open sharing of high-resolution data
from Microwave links, Radar, and Gauges (OpenMRG) will facilitate research on microwave-based environmental
monitoring using CMLs, and support the development of multi-sensor merging algorithms.



## 1 Introduction

Monitoring rainfall is of critical importance to society in many different ways, e.g. for design of infrastructure and post-analysis of rainfall-induced problems and disasters, for hydrological modelling and flood forecasting and not least for assessing climate variability and change. Heavy precipitation is already intensifying as a function of global warming, not

5    least in northern Europe (Masson-Delmotte et al., 2021), and this intensification is expected to continue (e.g. Olsson et al., 2017). Monitoring is however highly challenging due to the extreme spatiotemporal variability of rainfall. Rain gauges are still considered the most reliable, even though these are also known to give erroneous measurements (Habib et al., 2001; Sieck et al., 2007). Additionally, gauge networks are typically small or sparse (or both) and will therefore inevitably miss spatial information outside the network and in between the gauges.

In the last decades, weather radar has emerged as a key complementary rainfall observation technique. Radar pulses are transmitted in all directions and rainfall intensity may be estimated from their echoes. The key advantage is full spatial coverage in a circle around the radar. However, rainfall intensity is uncertain due to a range of error sources, e.g. clutter, attenuation and anomalous propagation (Battan, 1973; Zawadzki, 1984; van de Beek et al., 2016). An illustration of error

15    sources can be found in Figure 1. Furthermore, in a sense weather radar has a similar type of limitation as gauge networks, either covering a large area with low spatial resolution or a small area with a high resolution. There is a need to explore also other techniques for rainfall monitoring, not least in developing countries where gauge networks are often insufficient and expensive radar systems are scarce.

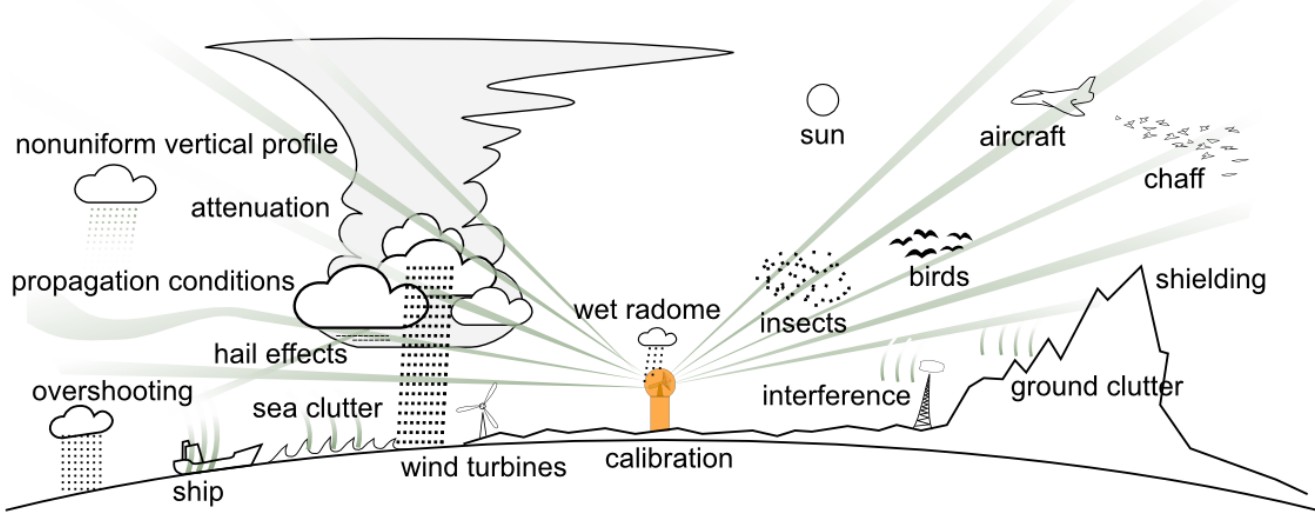

20    **Figure 1. Example of error sources effecting weather radar measurements. Figure by Markus Peura, Finnish Meteorological Institute (FMI), used with permission.**



The concept of using commercial microwave links (CML) in operational telecommunication networks for monitoring of rainfall and other environmental variables has been explored since about 15 years (Messer et al., 2006; Leijnse et al., 2007). The methodology to measure rainfall using microwave links exploits power attenuation of an electromagnetic wave when it propagates through water drops, and it is today well established that radio signals above 10 GHz are sensitive to rain and the

resulting power attenuation increases as frequency increases (ITU, 2005; Bao et al., 2017). Even if the conversion from attenuation to spatio-temporal rainfall fields is not trivial and requires a number of processing steps, estimating rainfall based on microwave links is a highly promising complement to gauges and radars, not least considering the very high resolutions in space and time attainable through the existing infrastructure, and their widespread operational use globally (Messer et al., 2006; Overeem et al., 2016; Fencl et al., 2015; Graf et al., 2021; Doumounia et al., 2014; Nebuloni et al., 2022; van de Beek

et al., 2020; Ericsson, 2018).

A major constraint for research on CMLs in operational networks concern data access (Chwala and Kunstmann, 2019). CML data are typically only available to mobile network operators and research groups they collaborate with. Given the potential of CMLs to act as opportunistic rainfall sensors, the data are however of general scientific interest, which is underlined by

the recently established OPENSENSE network (https://opensenseaction.eu/; https://www.cost.eu/actions/CA20136/). A few research groups have been able to share some CML data openly. Špačková et al. (2021) provided a 1-year data set for one dual-polarized microwave link sampled every ~4 s, along with disdrometers, rain gauges and conventional weather variables in Switzerland. Fencl et al. (2020) and Fencl et al. (2021) shared data from six E-band CMLs sampled every ~10 s, along with conventional gauge measurements of rainfall, temperature and humidity during four and seven months respectively in

Czech Republic. Overeem (2019) presented aggregated signal strength statistics at 15 min and ± 1 dB resolution from ~2500 CMLs, and radar data covering the Netherlands. Habi (2020) shared data from two CMLs in Israel, and van Leth et al. (2018) provided data for three research links in the Netherlands.

In order to significantly extend the open CML database available for research and benchmarking, the main objective of this

paper is to make available a set of CML data collected during a pilot project in Sweden. It consists of transmitted and received signal strengths from an operational network of 364 CMLs around Gothenburg during the summer months of 2015. The data are provided at high temporal resolution (10 s) and with true coordinates. To enable comparative analyses and algorithm development, we also include data from a set of precipitation gauges, one meteorological station and weather radar. Together, this makes the OpenMRG data set unique, providing data for many operational CMLs at high temporal

resolution and with precise coordinates, along with gauge and radar data.

In Section 2, the origin and collection approach of the included data sets are described. In Section 3, we proceed to describe the methods applied to present and analyse the data. Subsequently, section 4 presents the results of the data collection and the characteristics of the data. After some discussion about known challenges and potential applications in Section 5,

information on data availability is given in Section 6 and some concluding remarks in Section 7. We hope with this effort
will encourage cross disciplinary research in different related fields, e.g. environmental monitoring, sensor networks, big
data analytics, machine learning and hydro-meteorological forecasting.

## 2 Study area and data collection

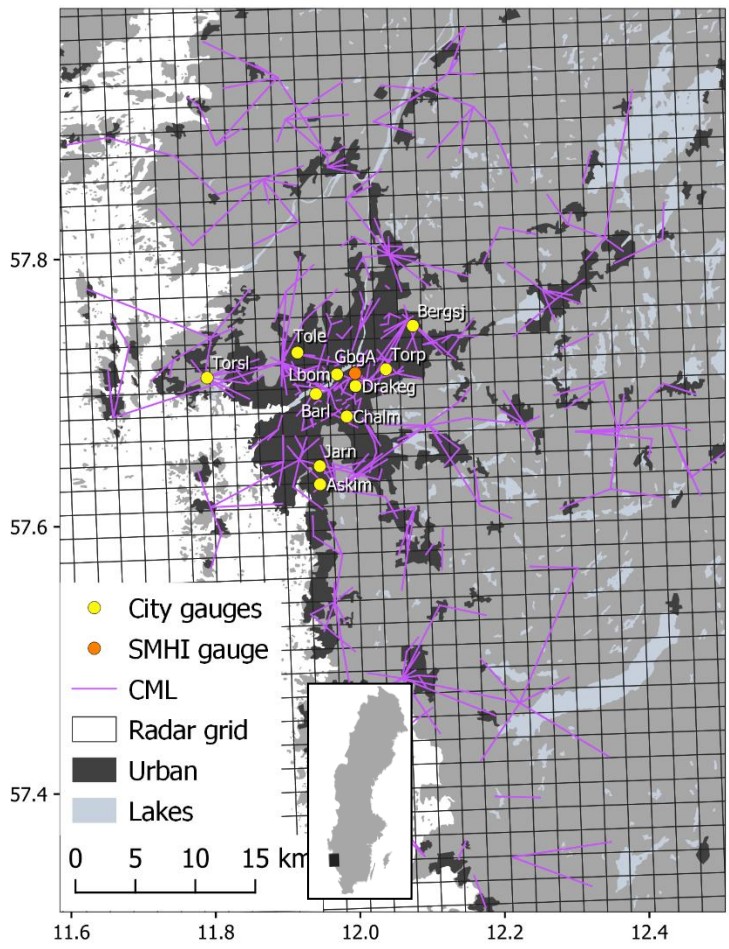

Figure 2. Map of Gothenburg, Sweden showing an overview of the data sets presented in this study. The inset map of Sweden shows the study area as a black square.

The data sets presented here stems from a pilot project involving the Swedish Meteorological and Hydrological Institute
(SMHI), the telecom company Ericsson and the mobile network operator Hi3G Sweden, which was initially reported at the
15[th] International Conference on Environmental Science and Technology (Bao et al., 2017; Andersson et al., 2017). The data
cover the city of Gothenburg, Sweden and its surroundings (Figure 2) for the period 1 June to 31 August 2015 (JJA 2015).



The OpenMRG data set consists of data from three types of rainfall sensors (CMLs, rainfall gauges, and weather radar) as well as standard meteorological parameters (temperature, humidity, pressure, wind).

## 2.1 Microwave links

For clarity, a set of terms are used to describe the CML data in this paper. A "node" is the location in which the antenna of a microwave radio is installed. A "sub-link" is a specific connection between two nodes, from a transmitting antenna (providing transmitted signal strength data, TX) to a receiving antenna (providing received signal strength data, RX). A "link" consists of all sub-links operating between two antennas. A link usually consists of two paired sub-links (one in each direction) but can consist of more, depending on design. Finally, a "hop" consists of all links operating between two nodes (e.g. one link operating at 18 GHz and another at 32 GHz). Figure 3 illustrates these terms.

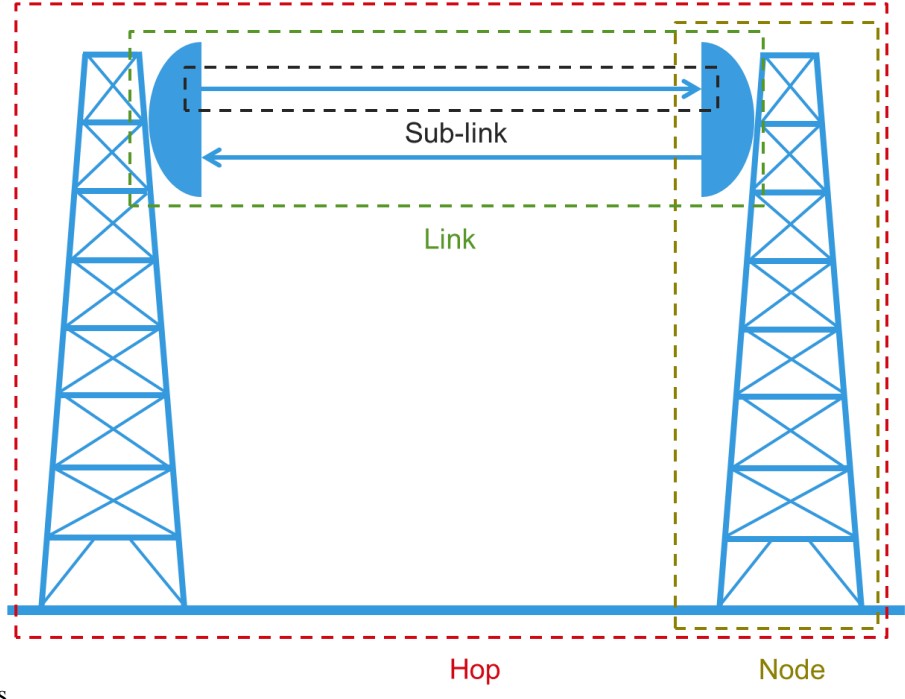

s

**Figure 3. Conceptual representation of a "node", "sub-link", "link" and "hop" respectively as used to describe CML features in this study.**

The CML data originate from a subset of the telecom operator Hi3G's operational microwave mobile backhaul network, specifically 728 sub-links operating using Ericsson MINI-LINK radios (Table 1; Ericsson, 2022; Morais, 2021). For each sub-link a set of metadata were compiled (ID, coordinates, frequency, polarization), which were extracted from the operator's network planning database. The network consists primarily of vertically polarized sub-links, operating at carrier frequencies between 7 GHz and 38 GHz, with a majority above 25 GHz (Figure 4a). Paired sub-links typically operate at similar frequencies (differing up to 1.5 GHz in the forward and reverse direction), which is the reason for the pairs in the





histogram in Figure 4a. The path lengths of the links vary between 100m and 15km, with a median of approximately 2km (Figure 4b). Higher frequency sub-links are typically installed at shorter distances (Figure 4c) to ensure robust operations despite increasing rainfall attenuation. The antenna diameters vary between 20cm to 1.2m, with corresponding antenna gain ranging between 31 dBi and 47 dBi. In general, antenna gain increases as antenna size and carrier frequency increase. The antennas were installed at 30m above ground on average.

**Table 1. General characteristics of the microwave link network**

| Characteristic | Value |
| --- | --- |
| Number of sub-links | 728 |
| Number of links | 364 |
| Number of hops | 363 |
| Number of nodes | 418 |
| Distance monitored by the links | 1 041 km |
| Area monitored by the links[*] | 2 800 km$^2$ |
| Temporal sampling resolution | 10 seconds |
| TX sampling resolution (quantization) | 1 dB |
| RX sampling resolution (quantization) | 0.3 dB |
| Adaptive Transmit Power Control (ATPC) | Disabled |

[*]Approximated by a convex hull around the CML network

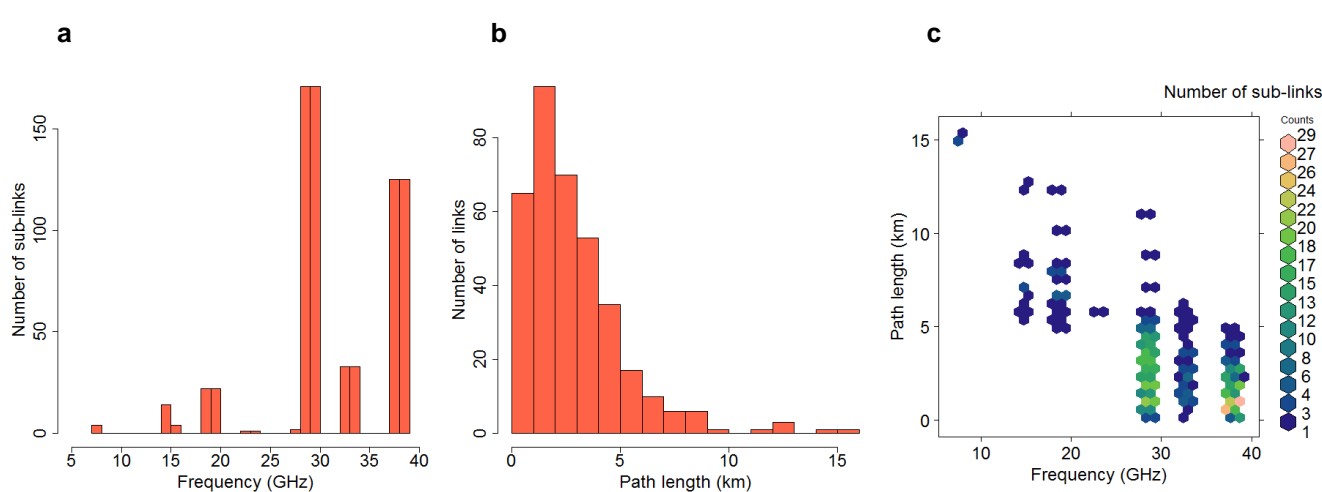

**Figure 4.** CML network characteristics: (a) distribution of carrier frequencies, (b) distribution of path lengths, and (c) relationship between path length, frequency and number of sub-links.

The CML network in Gothenburg has a star-shaped topology, i.e. with several links originating from the same node (Figure 5). Relatively short and higher-frequency links are more prevalent in the city centre, while longer low-frequency links are more common in the peripheral areas (Figure 5a). The network density is highest in the heart of Gothenburg, with at least 10 sub-links per km² (Figure 5b). This can be advantageous from several perspectives, including accuracy (using multiple

5    crossing links of varying frequencies to estimate rainfall), resolution (providing spatial information at sub-km resolution), and operational robustness (rainfall can be estimated even if some links are non-operational). The network density rapidly declines as distance from urban areas increase, with large areas being covered only by one link.

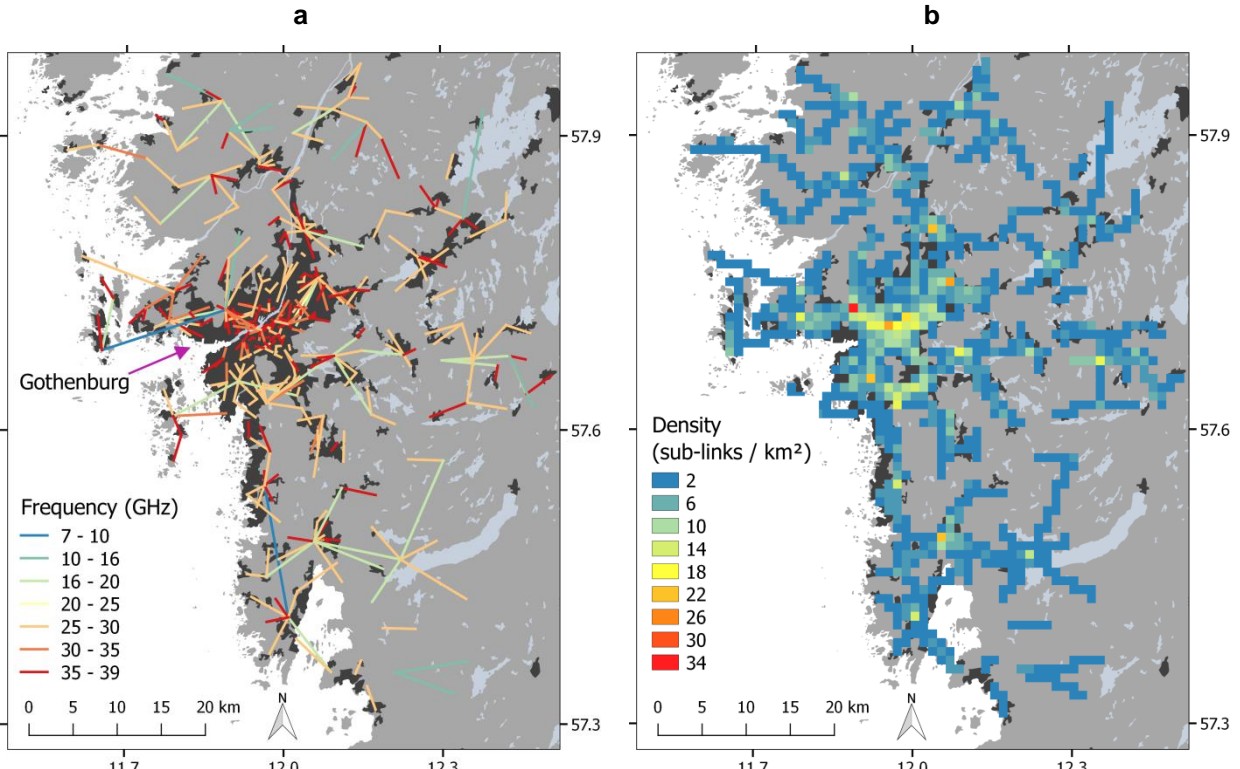

**Figure 5. Map of the microwave link network in Gothenburg, Sweden, showing (a) carrier frequency (b) and sub-link density.**

10    The collection of CML data from Hi3G's network was carried out by Ericsson. The CML monitoring system consisted of a data collection service (DC) and a data mediation service (DM). The DC was located inside the Hi3Gs network and utilized the Simple Network Management Protocol (SNMP) to sample and store TX and RX power levels every 10 seconds from a predefined set of nodes in the network. Every minute, the collected data were transferred to the DM, located at Ericsson. The DM compiled the data for further processing (e.g. mapping transmitting and receiving nodes, synchronizing time stamps,

15    assigning suitable IDs). In this process, values representing specific error codes or values that were outside of expected ranges were filtered out, specifically TX $\leq -99$, TX $= 255$, RX $\leq -99.9$, and RX $> -20$ were set to missing. Such values



appear due to radio timeout issues, when no signal was received, or due to other hardware impairments. Ericsson forwarded the data to SMHI in zipped TXT format. SMHI rounded the time steps to the nearest 10 s. This affected 67% of the time steps, that originally were sampled on second 59, 09, 19, 29, 39, 49 instead of 00, 10, 20, 30, 40, 50, respectively. Finally, SMHI calculated link lengths (Great Circle distance) and structured the data into a NetCDF file to facilitate further

processing. The collection system was designed in a flexible manner to enable varying temporal sampling resolution and link network, to enable potential extensions in the future.

## 2.2 Gauges

Two sets of observations from rainfall gauges are included in the OpenMRG data set. The first set is a time series of 15-min observations from station "GbgA" in the national network of automatic weather stations operated by SMHI (these data are

hereafter denoted "SMHI gauge"). The gauge, located in central Gothenburg (Figure 2), is of a weighing type (Geonor) with a 0.1 mm resolution which uses a precision vibrating wire transducer to weigh the precipitation collected. A thin layer of oil is added to impede any evaporation, which can essentially eliminate evaporation even during long periods without maintenance. The gauge is equipped with a wind screen in the form of metal or plastic plates that minimize precipitation losses, but still some wind-induced undercatch is likely to occur, e.g., caused by wind gusts and updrafts associated with

intense convective activity. Several other meteorological parameters are collected at the GbgA station, and we here include hourly observations of air temperature, relative humidity, air pressure, wind speed (average and gust) and wind direction. The observations have been quality controlled in the database system of SMHI.

The second set of rainfall observations comes from a gauge network operated by Gothenburg City (these data are hereafter

denoted "City network"). The network comprises 10 gauges well distributed over the entire city (Figure 2). Seven of these gauges are Geonor weighing gauges, i.e. the same type as the SMHI gauge. The other three are of a tipping-bucket type with a 0.2 mm resolution. Originally, the City network data were stored and provided with entries only for time steps with some recorded rainfall (i.e. without zeros or data quality flags). In the OpenMRG data set, we transformed all observations from the City network to complete 1-min time series by accumulating the recorded volumes from second 01 in the preceding

minute until second 00 of the minute given in the timestamp. Some known errors in the data were also removed, but the data did not go through the same rigorous quality control procedure as the SMHI gauge data.

## 2.3 Radar

The included radar data set is a subset of one of the operational NORDRAD composite radar products. NORDRAD was the operational radar product at the time for Sweden and the composite selected is based only on Swedish radars. This subset

covers the Gothenburg area with an extent of around 10km beyond the maximum extent of the microwave links. The projection parameters of the radar data can be found in Table 2. NORDRAD (Carlsson, 1995) is a set of operational products that are created within a close collaboration between different countries in the Nordic–Baltic region and more information on





NORDRAD can be found in Berg et al. (2016). This composite is created from all available radars at the lowest elevation (0.5°). The Gothenburg area is covered by three Swedish radars during this period: Vara, Karlskrona and Vilebo. Vara is the closest radar at 78 km. The other two are backups in case Vara does not provide data, but they cover only part of the domain as the other radars are at their extreme scan ranges. Details of the radar can be found in Table 3. More details of the Swedish

radars can be found in Norin (2015). The radar composite is also gauge-corrected by a distance dependent gauge–radar ratio estimation (Michelson and Koistinen, 2000).

The data contain radar reflectivity in pseudo-dBZ, meaning dBZ converted to values between 0 and 255 for efficient storage (255 represents missing data). The data values can be converted back to actual dBZ values with Eq. (1).

$$dBZ = 0.4 * value - 30 \qquad (1)$$

Equation 2 is needed for conversion of the dBZ to reflectivity Z:

$$Z = 10^{(\frac{dBZ}{10})} \qquad (2)$$

Finally, the rainfall intensity, R (mm h$^{-1}$), can be found from the following Z-R power-law relation, Eq. (3):

$$Z = aR^b \qquad (3)$$

where $a$ and $b$ describe the power-law coefficient and exponent. The coefficient $a$ is 200 and the exponent $b$ is 1.6 for a

standard Marshall–Palmer equation (Marshall et al., 1955). SMHI employs a slightly different exponent value during summer, i.e. $b$=1.5. The SMHI exponent value is used for the calculations represented in this paper and also recommended to use for validation purposes.

**Table 2. Metadata of the radar data set with a 5-minute temporal resolution. Here x- and y-scale represents the number of grid**
**point in each direction, and x- and y-size the spatial resolution of each grid cell. The upper left (UL) and lower right (LR) latitude (lat) and longitude (lon) coordinates are provided by the final four parameters. The projection is given as a "proj" string (https://proj.org/)**

| Component | Value |
| --- | --- |
| Projection | +proj=stere +lat_ts=60 +ellps=bessel +lon_0=14 +lat_0=90 |
| Xscale | 37 |
| Yscale | 48 |
| Xsize | 2000m |
| Ysize | 2000m |
| UL_lat | 58.0485 |
| UL_lon | 11.3953 |
| LR_lat | 57.2193 |
| LR_lon | 12.6739 |





**Table 3. Characteristics of radars contributing to the NORDRAD composite.**

| Radar | Lat | Lon | Antenna Height | Distance to station GbgA | Scan height above station GbgA | Frequency |
|---|---|---|---|---|---|---|
| Vara | 58.2556 | 12.826 | 164 m | 78 km | ~1.2 km | 5.605 GHz |
| Vilebo | 58.1059 | 15.9363 | 223 m | 237 km | ~5.6 km | 5.605 GHz |
| Karlskrona | 56.2955 | 15.6102 | 132 m | 270 km | ~6.8 km | 5.605 GHz |

## 3 Methods

The methods employed in this paper focus on presenting the characteristics of the data and the performance of the data collection process. As the gauge data include in the OpenMRG data set are intended to represent "ground truth", a limited
comparative description is included to highlight key characteristics and some differences. No further in-depth analyses are included since the main objective of the paper is to share data.

Two approaches were applied to assess the reliability of the data collection. Firstly, we calculated the overall number and percent of time steps for which data were successfully collected for at least one sub-link, rain gauge or radar pixel compared
with the total number of 10-second time steps in the study period. Secondly, we calculated the data collection hit rate for every sub-link, i.e. the percent and number of time steps with valid data for each the specific sub-link relative to all 10-second time steps in the study period. For TX, we also analysed the stability of the signal by calculating the range observed across the valid time steps for every sub-link.

Standard descriptive statistics (minimum, maximum, mean, standard deviation) were derived for each variable, and histograms plotted to show the distribution of the data. Moreover, the rain gauge observations were characterized in terms of the type of descriptive statistics that are commonly used in analyses of high-resolution rainfall observations:

- Total rainfall, *PTOT*: Total accumulated rainfall over the period (mm).
- Wet fraction, $WF^{ap}$: The fraction of all available time steps having a rainfall amount $p \geq 0.1$ mm at aggregation
period *ap* (min).
- Standard deviation, $STD^{ap}$: Standard deviation of the wet fraction depths (i.e. from all time steps with $p \geq 0.1$ mm) at aggregation period *ap* (min).
- Maximum rainfall, $PMAX^{ap}$: The maximum rainfall (mm) at aggregation period *ap* (min).

The last three statistics were calculated for aggregation periods (*ap*) 15 min and 1440 min (1 day). In these calculations, the
1-min observations from the City network were firstly aggregated into 15-min time steps, to correspond with the SMHI gauge.



To get a general overview of the rainfall events that occurred during the study period (JJA 2015), the gauge observations were plotted as time series as well as cumulative sums. We then analyse a high-intensity event to better understand the correspondence between the gauge, CML and radar data sets. RX time series were plotted along with gauge observations as a means of quality control by checking whether the expected behaviour – decreasing RX levels with increasing rainfall intensity – could be observed. Similarly, the time series of the radar pixel overlying the Torslanda gauge and a set of radar-based maps were plotted to illustrate its correspondence with the gauge and CML data.

## 4 Results

In this section we present the results of the data collection, and provide some fundamental characteristics of the data. We then plot the rainfall dynamics in the study area during the 2015 summer, and illustrate the correspondence between the different sensor observations through an intense rainfall event.

### 4.1 Data collection performance

The collection of data from the CMLs, gauges and radars was generally very successful (Table 4). Overall, 99.99% of the time steps were monitored by at least one sub-link in the CML network. For 93% of the sub-links, the data collection was successful ≥ 99% of the time (Figure 6a). Of the 728 sub-links, 16 lack either RX or TX data for the entire period. Typically, this is a result of failure to access specific nodes (e.g. because of maintenance), and it always affected the same links (i.e. 8 links in total). The sporadic missing data at the other sub-links can be caused by a variety of sources, e.g. blockages of the signal to the extent that no received signal was recorded, or due to radio timeout issues. The transmitted signal strength (TX) was constant for a majority of the sub-links (Figure 6b), because the network was operated without Adaptive Transmit Power Control (ATPC). This stability of the TX signal facilitates further analysis, since it becomes less sensitive to TX fluctuations and quantization error.

**Table 4. Overview of data collection results**

| Characteristic | CML[a] | SMHI gauge[b] | Radar |
|---|---|---|---|
| Potential number of samples within time period | 794 887 | 8 832 | 26 496 |
| Number (percent) of time steps with data for at least one sub-link/gauge/radar pixel | 794 836 (99.994%) | 100% | 26 381 (99.6%) |
| Number (percent) of time steps without any data | 51 (0.006%) | 0% | 115 (0.4%) |
| Total number of measurements | 560 918 042 | 8 832 | 46 617 762 |

[a] For CML, the values refer to valid pairs of RX&TX values. [b] Values represent the precipitation gauge at station GbgA.

Earth System
Science
Data

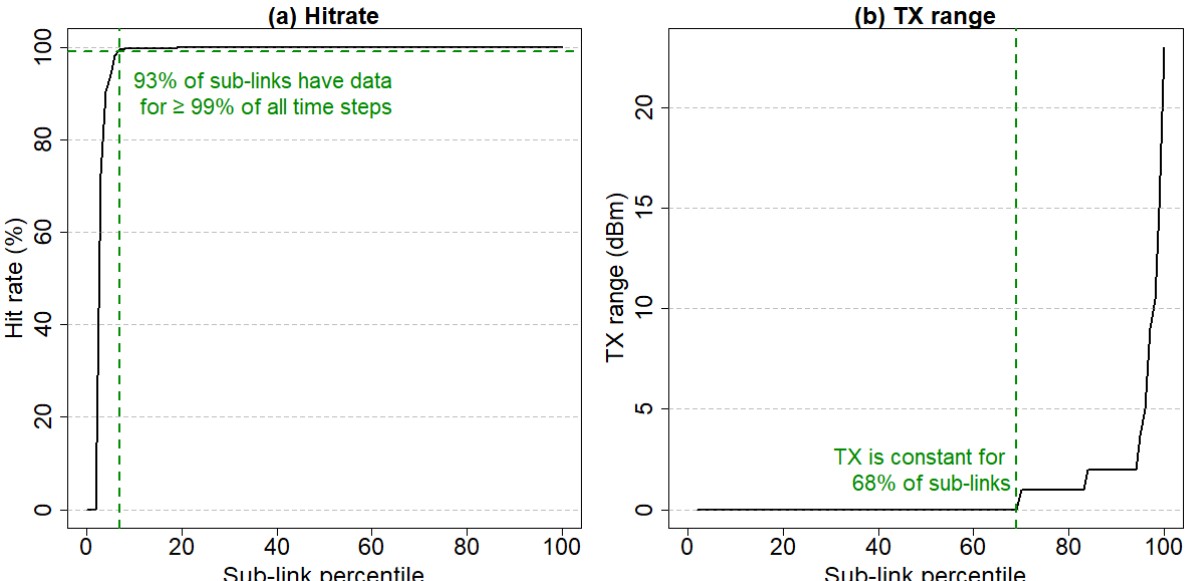

**Figure 6. CML data collection performance. (a) Distribution of the data collection hit rate (% of time steps with valid RX and TX measurements) across all sub-links. 93% of sub-links have both RX and TX data for ≥ 99% of all time steps in the sampling period. (b) Stability of the transmitted signal strength (TX) across all sub-links. TX does not vary at all for a majority of the sub-links (68%).**

No negative impacts on the CML network operations were observed due to the data collection. The collection load was approximately 10B s$^{-1}$ per node on average, and collection from the entire network scaled linearly with the number of nodes sampled (418×10B s$^{-1}$ = 4180 B s$^{-1}$), which is very small compared with the total network capacity that is often in the order of 100 Mb s$^{-1}$ or more. The data could also be collected with a very small time lag. The data collection step took only 1–3 seconds. By design, the system then waited until the entire minute was collected before passing on the data to archive storage and further processing. Overall this led to just over a minute in lag between observed event and data available, which is faster than most conventional rainfall monitoring systems. It can clearly be optimized further, but these results already indicate that CMLs are suitable for real-time applications (as demonstrated at https://www.smhi.se/memo).

The weight-based SMHI gauge GbgA recorded data for 99.95% of the 15-min intervals in the study period (Table 4), i.e. providing a near-complete record of precipitation events at that location. The completeness of the data collection for the City gauges is not known since they only store data when rainfall occur (i.e. it is not possible to distinguish zero from missing data). However, the similarity of their records to the SMHI gauge (see below) indicates that the completeness of the City gauge records is likely high. The radar data also display a high completeness, with only about 0.4% of the potentially available time steps missing data (Table 4). All in all, the data collection was successful providing near-complete time series for the CMLs, gauges and radars available in the area.




## 4.2 Descriptive statistics

Table 5, Figure 7 and Figure 8 present the distribution and descriptive statistics of the collected CML and radar data. The TX measurements have a relatively even distribution, from −6 to 18 dBm, except for a peak at 12 dBm. This reflects the design and authorized operating conditions of the CML network. The RX measurements are concentrated around the mean, which likely represents the prevalence of dry stable conditions for the vast majority of time steps. Still, there are some significant deviations from the mean especially at the left tail (down to −93.7 dBm), which is where the information relevant for rainfall estimation lies. To illustrate this, we calculated the difference between the median RX and the minimum RX for each sub-link. For the OpenMRG data set, this yields a dynamic RX range of about 30 dB on average (varying between 0 and 54.6 for individual sub-links). This variability stems from a range of factors, of which signal attenuation due to rainfall is one of the most important.

Figure 8 and Table 5 provides basic statistics and a histogram of the reflectivity distribution in dBZ for the entire radar data set (all data points contained in the data set). As can be seen, there are few reflectivities that are above 40 dBZ, which is a common threshold for convective precipitation (Steiner et al., 1995). It is also interesting to note the small peak around −15 dBZ which might be caused by clutter. The minimum value in Table 5 of −30 dBZ represents the lower limit of the radar and should be considered as zero precipitation. The low mean illustrates the high amount zero precipitation.

**Table 5. Descriptive statistics of the collected CML and radar data.**

| Sensor | Variable | Minimum | Maximum | Mean | Standard Deviation |
|---|---|---|---|---|---|
| CML | Received signal strength (RX, dBm) | −93.7 | −25.3 | −44.3 | 3.2 |
| CML | Transmitted signal strength (TX, dBm) | −10 | 26 | 8.5 | 6.2 |
| Radar | Reflectivity (dBZ) | −30.0 | 58.8 | −25.4 | 13.9 |

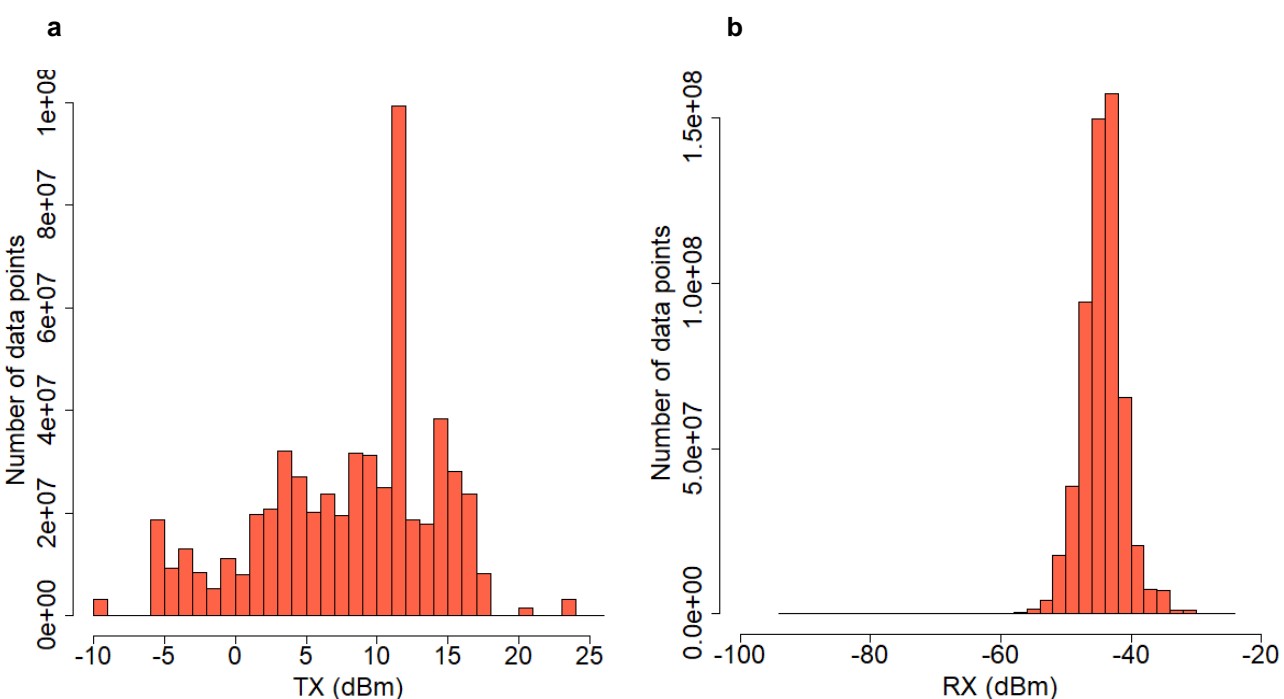

**Figure 7. Histograms of (a) transmitted signal strength (TX), and (b) received signal strength (RX).**

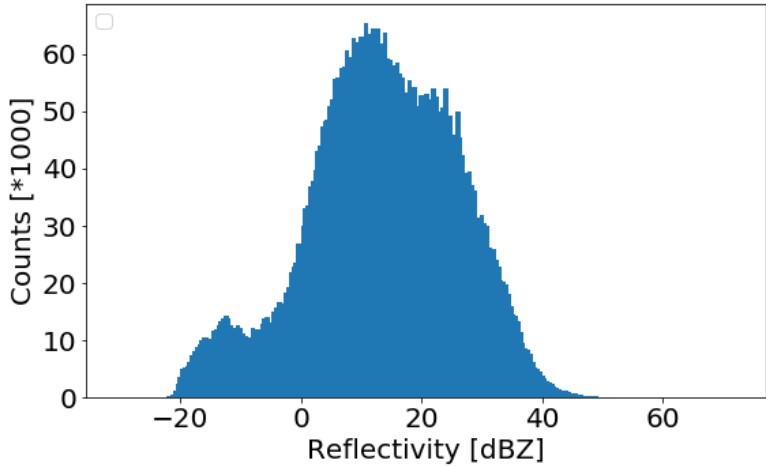

**Figure 8. Histogram of radar reflectivity (dBZ). Note, for radar the counts of the minimum values at −30 dBZ (representing 0 mm h−1, 41 691 984 counts) were removed.**

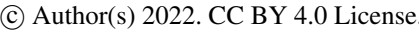

Table 6 provides key descriptive statistics for all rain gauge observations. Based on a limited climatological analysis of observations since 1996 (not shown), it was found that the period under study (JJA 2015) overall well represents an "average summer" with respect to rainfall in Gothenburg, although the maximum values (*PMAX*) are somewhat on the low side. Looking first at total rainfall during the period, *PTOT*, it is 256 mm in the SMHI gauge. The mean value from the weighing stations in the city network (263 mm) is very close to the SMHI gauge, indicating that the SMHI gauge well represents the city in aggregate terms, as also suggested in Figure 10. The tipping-bucket gauges, however, have a distinctly lower mean value (219 mm), which may be caused by e.g. shielding and excessive undercatch due to suboptimal placement, evaporation losses or occasional clogging. Much of the difference can be attributed to single periods with high-intensity rainfall affecting only parts of the city.

At accumulation period (*ap*) 15 min, the wet fraction (*WF*) in the SMHI gauge is 5.5%. Whereas the weighing gauges in the city network generally have a slightly higher *WF*, the tipping-bucket gauges' *WF* are distinctly lower. The highest *WF*-value among the tipping-bucket gauges (4.6%) is almost one percentage point lower than the lowest *WF*-value among the weighing gauges (5.4%). In the SMHI gauge $WF^{1440}$=48.9%, i.e. it rained on average every second day. Also, in this case, *WF* is higher for the weighing gauges in the city network and distinctly lower for the tipping-bucket gauges. In the city network, overall the *WF*-values at different accumulation periods are similarly ranked, i.e. a high value at *ap*=15 min implies a high value also at *ap*=1440 min, although minor deviations from this pattern exist.

In the SMHI gauge, the standard deviation *STD* ranges from 0.69 mm at 15 min to 7.23 mm at 1440 min accumulation period. These values are overall well matched by the City network gauges, regardless of type. In the SMHI gauge, the maximum value $PMAX^{15}$=7.1 mm and $PMAX^{1440}$=25.5 mm. Also, these values are overall well matched by the City network gauges. A distinct spatial variation in the seasonal maximum values is expected, particularly at short durations, considering the small-scale convective rainfall processes that are generally involved in generating these extremes.





**Table 6. Key rainfall statistics for the rain gauge observations during June–August 2015. For gauge locations, see Figure 2.**

| Network | Type | Name | PTOT (mm) | WF$^{ap}$ (%) | | STD$^{ap}$ (mm) | | PMAX$^{ap}$ (mm) | |
|---------|------|------|-----------|--------|----------|--------|----------|--------|----------|
| | | | | ap=15 | ap=1440 | ap=15 | ap=1440 | ap=15 | ap=1440 |
| **SMHI** | *Weighing* | GbgA | 256 | 5.5 | 48.9 | 0.69 | 7.23 | 7.1 | 25.5 |
| **City** | *Weighing* | Jarn | 237 | 5.9 | 53.3 | 0.59 | 7.02 | 4.9 | 28.8 |
| | | Torp | 271 | 6.6 | 55.4 | 0.59 | 7.43 | 4.6 | 27.4 |
| | | Bergsj | 301 | 6.6 | 57.6 | 0.76 | 7.78 | 6.9 | 25.2 |
| | | Torsl | 273 | 5.6 | 50.0 | 0.78 | 9.55 | 8.6 | 44.3 |
| | | Chalm | 277 | 6.3 | 62.0 | 0.73 | 7.73 | 8.7 | 31.3 |
| | | Tole | 251 | 5.9 | 54.3 | 0.65 | 7.91 | 6.8 | 36.6 |
| | | Barl | 232 | 5.4 | 50.0 | 0.73 | 6.98 | 8.9 | 26.6 |
| | | MEAN | 263 | 6.0 | 54.7 | 0.69 | 7.77 | 7.1 | 31.5 |
| | *Tipping-bucket* | Drakeg | 185 | 4.3 | 45.7 | 0.49 | 5.59 | 4.0 | 19.4 |
| | | Lbom | 228 | 4.6 | 46.7 | 0.61 | 6.60 | 5.4 | 23.2 |
| | | Askim | 245 | 4.2 | 42.4 | 0.81 | 7.52 | 8.4 | 30.8 |
| | | MEAN | 219 | 4.4 | 44.9 | 0.64 | 6.57 | 5.9 | 24.5 |

## 4.3 Rainfall dynamics during summer 2015 as observed by the gauges

Figure 9 shows time series plots of rainfall in the SMHI gauge during the study period (JJA 2015). The highest 15-min accumulations (up to 6–7 mm) were recorded on 28–29 July, but the highest daily accumulation (25.5 mm) on 17 June. Whereas the former event was dominated by isolated very high 15-min intensities, the daily maximum was caused by a nearly 10-h long low-intensity event. Several short high-intensity events occurred surpassing the national 1-minute cloudburst threshold of 1 mm (e.g. nine times at station Bergsj), although no events were recorded surpassing the national 1-hour cloudburst threshold (50 mm in one hour).



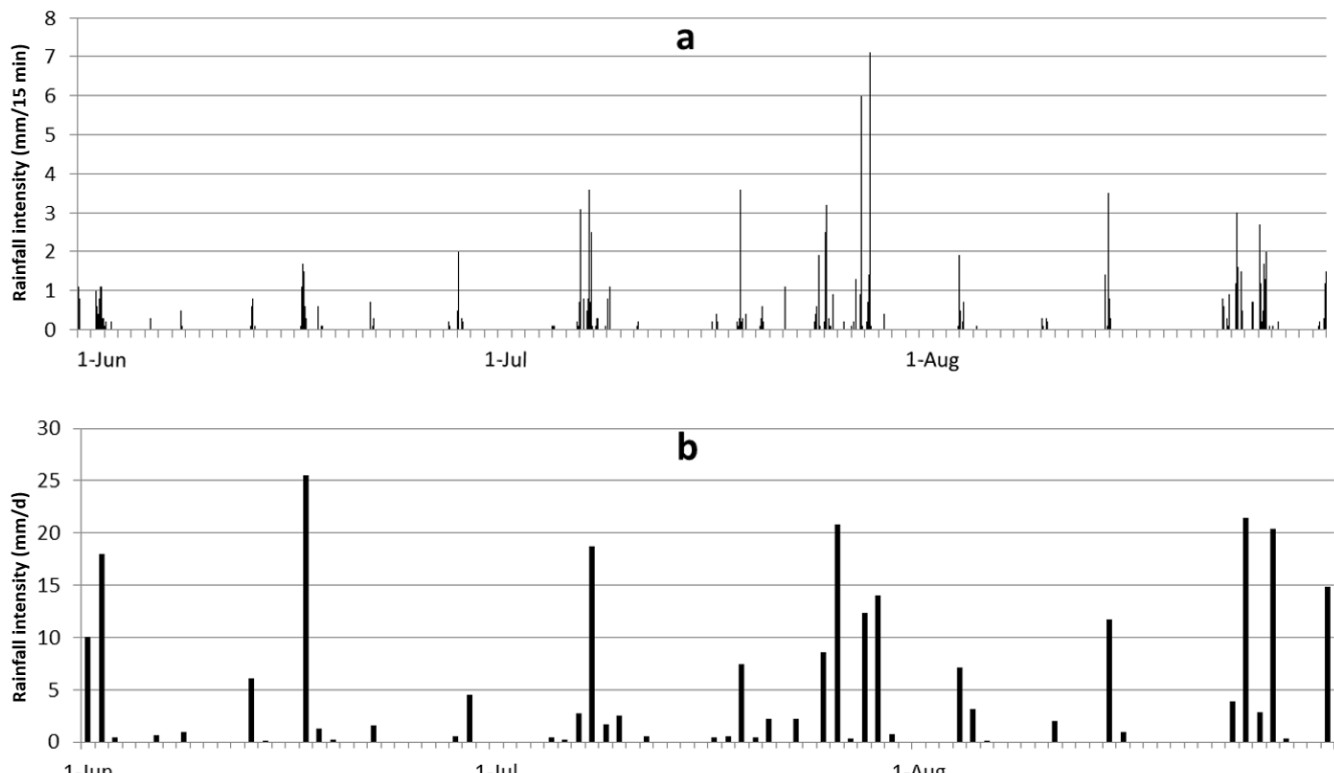

**Figure 9. Rainfall time series in the SMHI gauge at 15 min (a) and 1 day (b) temporal resolution.**

The cumulative sum during the period shows that the rainfall in the City network gauges (*i*) follows the pattern in the SMHI gauge well and (*ii*) are rather evenly distributed around the SMHI gauge (Figure 10). Differences in rainfall during two periods are responsible for most of the spread between the gauges; the first one on 1–2 June and the second one on 28–29 July. Both periods were characterized by large spatial variability with some gauges recording only a few mm and others up to 50 mm.



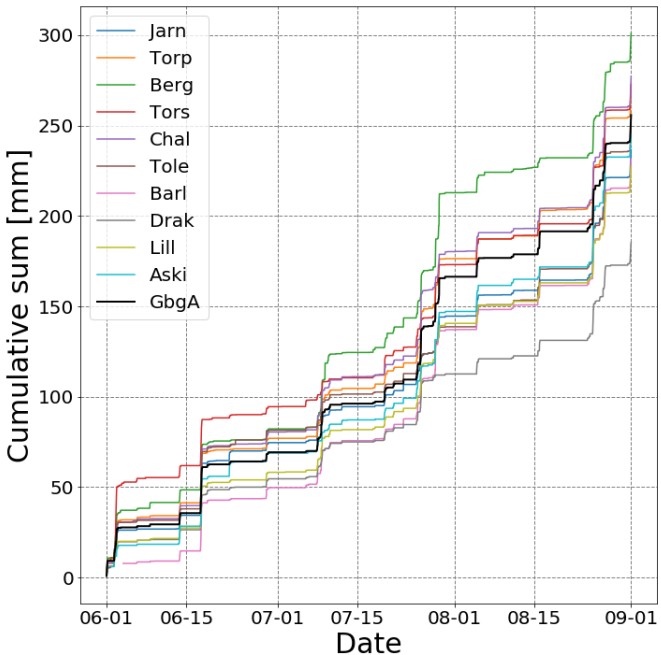

**Figure 10. Comparison of the SMHI gauge and the City network gauges. The figure shows the cumulative rainfall sum in each gauge over the study period.**

## 4.4 Case study: the high-intensity event on 28 July 2015

The intense rainfall event recorded on 28 July at Torslanda (Figure 2, Figure 12) serves to illustrate the strong correspondence between gauge and CML data. The event lasted for 2 hours with a peak 1-minute intensity of 1.1 mm min$^{-1}$ and a total accumulation of 18.5 mm. The expected behaviour is that received signal levels (RX) will decrease when rainfall intensity increases and vice versa (ITU, 2005). This was observed on many occasions, and is here illustrated during 28 July (Figure 11). The event also serves to illustrate how radar data is useful to map the overall spatial pattern of an event, but sometimes fail to represent the local details observed on the ground (Figure 13, Figure 11). Before the event (14:30 to 14:55 UTC), rainfall intensity at the gauge and radar is zero, and the RX range is small (around −46.5±0.5 dB). The first gauge record of the event is between 14:58 and 14:59, while the sub-links record the first RX decrease at 14:57:20. The small difference in time is likely caused by the movement of the storm coupled with the difference in geographic position and monitored area of the links and the gauge. At this time, the radar indicates zero rainfall at Torslanda, but also some rainfall a few km away. The peak intensity (1.1 mm min$^{-1}$) is recorded between 15:04 and 15:05 at the gauge, while the deepest RX fade is at 15:06:00. At this stage the RX levels have dropped significantly, about 27 dB down to −73.6 dBm. The intensity then goes down at the gauge, and RX levels follow suit by increasing again toward their starting values. During this peak the radar indicated that no rainfall occurred at the grid cell overlying Torslanda (and at adjacent grid cells), but also that some



higher-intensity rainfall occurred several km further north. The reason for the failure of the radar to capture the event at Torslanda is unknown, but it could be due to e.g. spatial misalignment (placing the peak too far north), low temporal sampling frequency (one snapshot every 5 minutes), overshoot (giving no echo at Torslanda), or attenuation between the radar and Torslanda (diminishing signal strength), Figure 1. At 15:22, the gauge again reports zero rainfall, while at the same time the RX recession is still ongoing (i.e. RX is still not back to the initial pre-event levels). This is likely an effect of wet-antenna attenuation as well as the difference in geographic position. Then follows a lighter shower peaking at 15:45 and observed by the gauge, the CMLs and the radar. At the end of this, the gauge displays a jerky behaviour with alternating records of 0 and 0.1 mm min$^{-1}$. This is probably a result of the measurement resolution of the gauge. The CML and radar data here provide a smoother representation, which is more realistic.

In general, the sub-links display similar temporal dynamics as the gauges, but there are also exceptions. For example, the last peak observed at Torslanda during this event (0.7 mm min$^{-1}$ at 16:22, Figure 11) resulted in a smaller RX drop compared with the same intensity during the first peak. One potential reason could be differences in the spatial alignment between the location of the most intense rainfall and the gauge and sub-link locations, respectively, for the two events. Another reason could be that the antenna was wetter during the first peak than during the second, resulting in less wet-antenna attenuation during the second peak. The open publication of this data set provides a good opportunity to further analyse the correspondence between RX dynamics, gauges and radar more comprehensively.

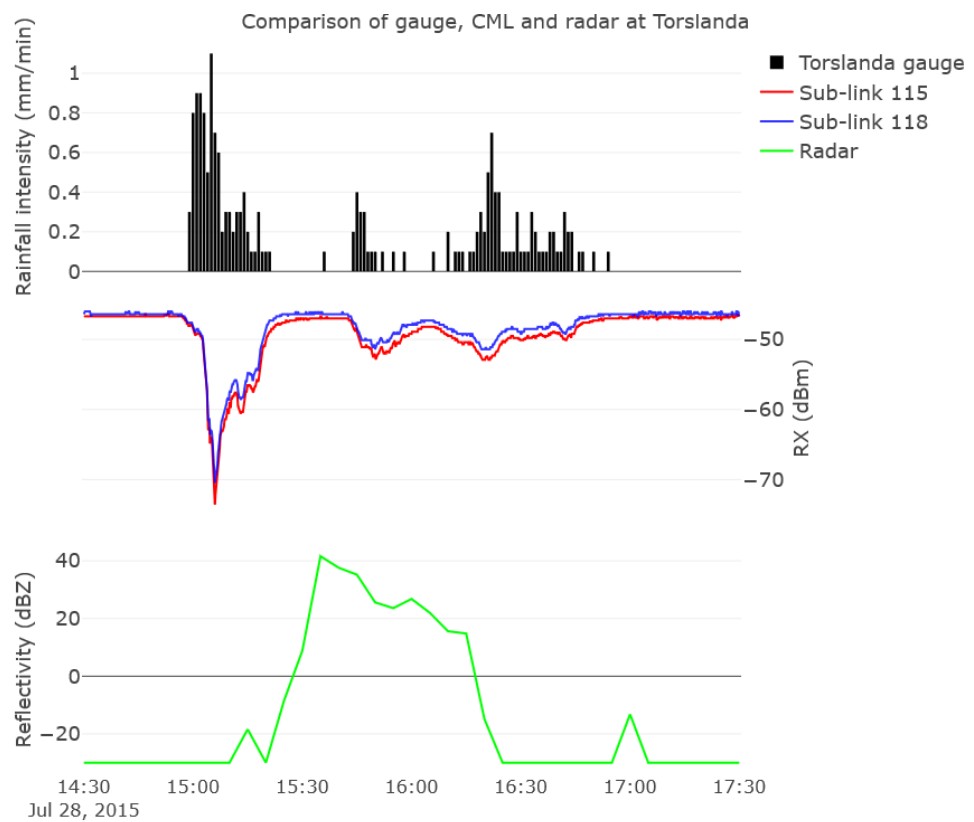

**Figure 11. An intense rainfall event at the Torslanda gauge on the 28 July 2015 (top), and the corresponding RX signal fade for a nearby microwave link (middle, link 10227, sub-links 115 and 118) as well as the reflectivity of the overlying radar pixel (bottom). The location of the gauge and link are shown in Figure 12. Sub-link 115 is vertically polarized, with frequency 28.23 GHz, covering a 1.8km distance. Sub-link 118 is identical except for operating at 29.24 GHz. An interactive version of this figure is available together with the data set (see section 6).**

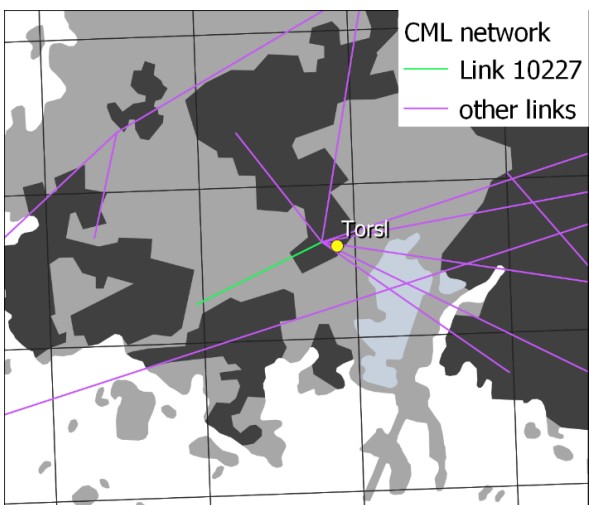

**Figure 12. Map of the Torslanda gauge (Torsl) and surrounding CMLs, with link 10227 highlighted, as used in Figure 11.**



**Figure 13. Estimated rainfall intensity based on the radar data at key time steps during 28 July 2015. The red circle indicates the location of the Torslanda gauge.**

## 5 Discussion

The above comparative description of rain gauge observations in Gothenburg during summer 2015 well illustrates the very high spatial and temporal small-scale variability of rainfall, also found in numerous other studies worldwide (e.g. Krajewski et al., 2003). Consequently, accurate observation requires very high resolutions in space and time and CML networks offer a unique possibility to perform such high-resolution observations. A key advantage is that the CML infrastructure is already in place: the number of operational CML hops in Sweden is more than 20 000, in Europe more than 400 000, and several millions globally (Ericsson, 2018).

The primary intended use of the OpenMRG data set is to allow a wider research community to develop and refine algorithms for rainfall retrieval from CML networks. A number of data processing challenges have been identified. The first set of





challenges concern separating rainfall information from other factors affecting the microwave signal, such as refractivity-induced multipath fading, wet-antenna attenuation, intermittent disturbances due to human activities (e.g. construction cranes, ferry crossings, maintenance), canopy foliage dynamics, antenna misalignments, and multipath fading over water bodies etc. (Bao et al., 2017). A second set of challenges concern estimating rainfall intensities accurately for links with
different frequencies, polarization, lengths, temporal resolution, signal quantization, unstable baseline conditions, and operating in different climatic, physiographic and management conditions. A third category of challenges concern integration of different links, and potentially also gauges, radars and other information sources into high-resolution maps and spatio-temporally complete data sets. Significant progress has been made by a number of research groups on many of these challenges over the past decade. Conceivably the extensive data included in the OpenMRG data set will make it possible to
further improve our capacity to meet these challenges, and to help scrutinize the general applicability of existing methods.

There are several applications of up-to-date high-resolution rainfall data. One key application is real-time mapping of current and past events, which are used e.g. in operational meteorological forecasting, model evaluation, and post-event insurance analyses. CML networks can be used to create near real-time high-resolution rainfall maps, as demonstrated at
https://www.smhi.se/memo. Another key potential application is hydrological modelling and forecasting, as demonstrated by e.g. Fencl et al. (2013). Frequent initialization of the hydrological model, i.e. model state update to reflect current hydrological conditions, is crucial for accurate performance (Hapuarachchi et al., 2011). Further, hydrological response to rainfall always has a delay, which may be days or weeks in large rural basins, but below 1 hour in small urban basins. Using the most recent rainfall observations is thus a very fundamental prerequisite for accurate hydrological forecasts. Considering
the particularly dense CML networks in cities, urban applications are of obvious significance, such as real-time control of sewer systems for quantitative and qualitative water management, but also rural hydrological forecasting in poorly or un-gauged regions is a motivating prospect. These possibilities are already being explored (e.g. in Stockholm, von Scherling et al., 2021), and the open sharing of OpenMRG aims to further such research and applications.

## 6 Data availability

The OpenMRG data set is openly available at https://doi.org/10.5281/zenodo.6673751 (Andersson et al., 2022) under the Creative Commons Attribution Share Alike 4.0 licence. More details on the data structure and files the are provided in the data repository. The OpenMRG data covers the period June–August 2015. More data for the GbgA station is available at https://opendata.smhi.se/ (specifically, https://opendata-download-metobs.smhi.se/explore, last accessed 2022–06–21). In addition to the data we also include scripts to read the data, an interactive version of Figure 11, and a radar animation
covering the 2015 event at Torslanda (Figure 13).

Earth System
Science
Data

## 7 Concluding remarks

Technical development, e.g. information technology and digitalization, occasionally brings unexpected and unintentional opportunities for additional benefits. Rainfall monitoring through processing of microwave network signals is a good example of such an opportunity. Although development remains in terms of both data handling and conversion algorithms,
the step to large-scale and widespread operational monitoring using CMLs is relatively small from a technical perspective, since the infrastructure is essentially already in place. Rather, the main challenge to overcome concerns enabling access to CML data. Key to this endeavour is to develop mutually beneficial collaborations and viable business models involving key actors, such as mobile network operators, CML manufacturers, hydro-meteorological agencies, municipalities, and researchers. It is our hope that rainfall monitoring by CML networks will soon reach widespread operational implementation
and thereby provide a distinct added value in a sustainable and climate-proof society.

*Author contribution*

JCMA conceived the idea and secured that the CML and City network data could be shared. JH collected the CML data, and JCMA prepared and analysed the CML data. CZB gathered, prepared and analysed the radar data. CZB and JO gathered, prepared and analysed the gauge data. JCMA, JO, CZB, and JH acquired the funding and wrote the manuscript.

*Competing interests*

The authors declare that they have no conflict of interest.

*Acknowledgements*

We acknowledge the indispensable support for this work by Hi3G Access AB (providing access to the CML network, through Håkan Andersson and Håkan Snis), and Göteborgs Stad – Kretslopp och Vatten (providing the City network data,
through Jonas Persson). This work was carried out in the context of the projects MEMO (financed by Vinnova, dnr: 2017-03297), FutureCityFlow (financed by Vinnova, dnr: 2019-04701), and Urban skyfallsinformation (financed by the Swedish Ministry of the Environment and Energy, grant 1:10 for climate adaptation). We also thank Lei Bao, Anna Jacobsson, Christina Larsson, Mohamed Mustafa, Mikael Riedel, Johan Selin, Victor Näslund, and Johan Thuresson for valuable technical contributions and discussion.



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
