# Peer review of "OpenMRG: Open data from Microwave links, Radar, and Gauges for rainfall quantification in Gothenburg, Sweden"

_Earth System Science Data, 2022_

## Author Response (AR1)

**Reply to Referee #1 (RC1, https://doi.org/10.5194/essd-2022-221-RC1)**

Dear Referee #1,

Thank you for your review of our manuscript, your encouragement on its quality and importance, and your comments. Please find below our response to your comments.

**RC1, Comment 1**: "My only concern about the paper and the provided data is the access and direct usability of the data. The authors provide access to raw CML and radar data. For the radar data, the paper presents equations and standard parameters to convert the raw reflectivity data into rain rates. However, this is not the case for the CML data. Either the authors could provide CML data that has already converted into rain rates (such as a rainfall product) or provide information on algorithms or a literature review on how to perform the conversion. Also, methods on e.g. bias-adjustment of both CML and radar data could be provided. By both supplying the raw data as well as directly applicable rainfall data, potential users could 1) use the data directly, e.g. in research in hydrological modeling, or 2) research further into conversion, bias adjustment and comparisons with other raw data conversion methods. Including both types of data would increase the number of potential users and create a basis for fundamental CML and radar research as well as an applicational leg."

**Authors' response:** It is true that there is currently an unbalance between the radar and CML descriptions in this sense. We will revise the manuscript with a short description on how raw CML data can be converted rainfall intensity (a basic algorithm). However, we prefer not to add rainfall time series derived from the CML data. The main reason is that CML processing can be done in several different ways – each with their pros and cons – and it entails making assumptions and interpretations, which need to be described and evaluated before use. This is beyond the scope of our manuscript. Our main ambition here is to provide raw data from several sensors on which different precipitation algorithms could be developed and/or evaluated (based e.g. only on CML or on several sensor types). This is also in line with the comment from referee #2.

**Authors' changes:** We added a paragraph describing a basic workflow for how to derive rainfall from CML data (page 9, line 11-24).

**Reply to Referee #2 (RC2, https://doi.org/10.5194/essd-2022-221-RC2)**

Dear Referee #2,

Thank you for your review of our manuscript and data set, your recognition of its significance and your many helpful comments, suggestions, and corrections. Please find below our responses.

**RC2, Comment 1:** "Naming convention of TX and RX: While I have seen the term TX and RX being used for CML data, using TSL and RSL (transmitted signal level) and (received signal level) is much more explicit. Why was TX and RX chosen and would it be possible to change to TSL and RSL?"

**Authors' response:** TX and RX was chosen because it is the most common abbreviation used within the telecommunication industry. Several terms are though used interchangeably: transmitted signal power, transmitted signal strength, and transmitted signal level. We can change to TSL and RSL in the revised manuscript to be more explicit.

**Authors' changes:** We changed the name to TSL and RSL in the Zenodo repository (cml.nc, example_read_cml.nc.py, Torslanda-interactive-time-series.zip), the figures (e.g. figure 6), and the manuscript (e.g. page 5 line 6 onwards). We also included an explanation of the link between these terms in section 2.1.

**RC2, Comment 2:** "Rainfall data from CMLs: I agree with reviewer #1 that it would be beneficial to add information on how to process the raw CML data into rain rates. I am, however, undecided whether or not the addition of processed CML rain rates is required or even a good idea. I see the presented dataset as the basis for the validation of existing and new CML processing techniques and not as a tool to use CML rainfall data in an hydrological application, because (at least in my opinion) expert knowledge on CML processing is still required to interpret certain misbehaviour of CML rain rates caused by e.g. erratic behaviour of the CML data or by wet antenna attenuation. If the authors do not yet have an optimized processing for the presented CML data, it would be too much effort to produce it for adding it here. But, a small section with info regarding CML data processing methods and code, would be beneficial for sure."

**Authors' response:** We agree. We will revise the manuscript with a short description on CML data processing methods.

**Authors' changes:** We added a paragraph describing a basic workflow for how to derive rainfall from CML data and a link to a useful code package (page 9, line 11-24).

**RC2, Comment 3:** "For some of the city gauges the metadata entry for „type" und „unit" is switched"

**Authors' response:** Great that you noticed, we will fix it and update the metadata.

**Authors' changes:** We fixed the switched metadata entries and revised the file CityGauges-metadata.csv

**RC2, Comment 4:** "The proj string that is given in the paper is not available as attribute in the NetCDF. To do the reprojection of the coordinates I had to copy-paste it from the paper. That is easy, but all information required to work with the data should be available from the data alone. There is a lot of info on the projection in the NetCDF in the variable „crs", but I could not find something that I can directly use in my code for doing the projection."

**Authors' response:** The CRS variable contains all projection information. But, to simplify for data users we will also add the proj-string as an attribute.

**Authors' changes:** We added the proj-string to the global attributes in radar.nc.

**RC2, Comment 5:** "I strongly suggest to add the longitude and latitude grid to the radar data NetCDF. I took me some time and some internet search to do the reprojection from the provided x and y data to lon and lat. CML and gauge data also come with lon-lat coordinates. Hence, for comparison with

radar data, one has to generate the lon and lat grids. While this is doable, it is prone to errors and cumbersome (not the actual coding, but finding the solution, at least if you are not doing this on a daily basis) [...] The lon_grid and lat_grid can be added to the NetCDF as variable along the (y, x) dimension. The variable with the pseudo-dBZ data should then have an attribute "coordinates": "longitudes latitudes", assuming that the lon and lat grid variable name is „longitudes" and „latitudes". I am not an expert on this, but I think it is also good to add an attribute „grid_mapping": „crs" to the pseudo-dBZ variable so that any NetCDF viewer will correctly plot the data on a map. As far as I can see the already existing variable „crs" has a lot of projection info. I do not know if this info is in line with the CF conventions (cfconventions.org), though.".

**Authors' response:** The projection contained in the radar data NetCDF is the same as the one used in the original NORDRAD radar data set and was chosen to reduce the chance of errors. We tested the NetCDF and it was projected without issues in QGIS as it is now. However, we will add additional lat and lon coordinate variables to the NetCDF to make things easier for users that prefer this.

**Authors' changes:** We added the lat and lon fields as variables to the NetCDF file where the coordinates are in the centre of the grid cells.

**RC2, Comment 6:** "The usage of pseudo-dBZ is not required to have small NetCDF file size. NetCDF supports these types of conversion automatically when reading or writing. One can set the encoding attributes "scale_factor", "add_offset" and specify the "datatype" of the variable (see http://cfconventions.org/cf-conventions/cfconventions.html#_reduction_of_dataset_size). This is not an urgent issue, but if the radar NetCDF is redone because the lon and lat grids are added, this could also be improved."

**Authors' response:** Thank you for this comment, we were not aware that this could be automatically linked in a NetCDF. We will look into adding this.

**Authors' changes:** We changed gain and offset to the 'scale_factor' and 'add_offset' attributes as suggested. This will result in the data being read as dBZ automatically in all software supporting it (e.g. Python, R). We modified the manuscript slightly to make this clearer.

**RC2, Comment 7:** "P2 L15: I can somehow understand the reasoning behind the analogy of the limitation of weather radar and gauges regarding the tradeoff between coverage and resolution. However, the resolution at which weather radar provides rainfall information can be improved, and has already been improved, by more advanced scanning techniques that allow faster sweep times. For rain gauges the only option to increase resolution, in the sense of observability of spatial gradients of rainfall fields, is to place more gauges. „Resolution" is also a vague term here, since it is well defined for radar observations, but not for observations of rain gauge networks. Hence, the authors might want to reconsider this sentence. Please note, that this is more of a subjective comment."

**Authors' response:** We will remove this sentence since this can indeed be a subjective interpretation and it is not essential for this publication.

**Authors' changes:** We removed this sentence in the Introduction (page 2, line 15).

**RC2, Comment 8:** "P3 L5: The reference Bao et al 2017 does not fit here. Besides the reference to ITU the authors could cite work from the 70s where the k-R relation has been studied in detail empirically and theoretically […]"

**Authors' response:** Ok thanks for the suggestions.

**Authors' changes:** We modified the sentence and added the references suggested (page 3, line 4).

**RC2, Comment 9:** "P3 L20: It should be made clearer here that the data from the Netherlands was not aggregated by the researchers, but rather is provided like this by the network operators from the performance reports that network management systems typically produce."

**Authors' response:** Ok, we will clarify this in the revised manuscript.

**Authors' changes:** We clarified this sentence (page 3, line 20).

**RC2, Comment 10:** "P5 Fig 3: This is an important figure to define these terms which, up to know, do lack a clear citable definition and have been used inconsistently. Hence, I suggest to improve the clarity of the figure. Some suggestions: The dashed lines of the boxes combined with the choice of colours makes it hard to distinguish the individual relevant parts. Maybe having the tower and antennas drawn in black and then using easily distinguishable colours for the boxes can improve that. Shaded solid coloured boxes might be better than the boxes with dashed lines, or just solid lines of coloured boxes if towers and antennas are in black. The text for „sub-link" should maybe be on the top and not between the two arrows. It would also be good to add a second „link" to make it clearer in the figure why the term „hop" is needed. It is also not clear to me why the „node" and „hop" comprise also the towers."

**Authors' response:** Thank you for the suggestions. We improved the figure in the revised manuscript based on these suggestions. The nodes include the tower/mast because there are often several links going in different directions from a certain node. All these links are sampled in the same request to the node. We illustrated this by adding also antennas in other directions. For the hop, we now draw it to finish in the middle of each mast, since it comprises all links going between two specific nodes (but not beyond). Coordinates represent the midpoint of a specific node. This means that the provided hop lengths in the data are between the midpoints of two nodes, and hence this way of drawing. The figure simplifies some aspects (e.g. the offset of the antenna from the midpoint of the mast), but stills sufficiently illustrates the key concepts.

**Authors' changes:** We revised Figure 3 to show the components more clearly, also considering colour blindness. We also revised the first paragraph in section 2.1 (page 5).

**RC2, Comment 11:** "P5 L14: Morais (2021) is an interesting reference that I was not aware of. But it should be made clearer here what specific part of this sentence it supports or provides more information on. Since this reference is a book, the effort to access the content is higher than for an online paper PDF. Hence, it would be good to be more specific here."

**Authors' response:** We understand, we will specify which parts of the book that are relevant.

**Authors' changes:** We expanded the sentence to point the reader to the relevant chapter in the book (page 5, line 18).

**RC2, Comment 12:** "P6 L3: Is the information on antenna size for each CML also available? If yes, would it be possible to add it to the open dataset? This might be important information regarding the investigations of the wet antenna effect."

**Authors' response:** Yes, it is available, we will add it.

**Authors' changes:** Added antenna diameters to cml_metadata.csv. Changed manuscript on page 5 line 20

**RC2, Comment 13:** "P7 L12: Is the sampling done at one fixed 10-second interval for all links, or are there individual sampling cycles or slight offsets for each CML?"

**Authors' response:** The sampling is not done exactly at the same time for all links. Sampling is done toward nodes, which happen both in parallel and in sequence, which means the exact sampling time varies slightly between nodes. In addition, there are slight temporal offsets between the sampling of TX and RX for a given sub-link, since two nodes must be sampled. By design, we allow a 7-second window for the sampling to finish. However, the analysis we have done on this indicate that all nodes are likely sampled within ±1 second for this network. The subsequent sampling of nodes follow the same sequence, to stay as close as possible to the target 10-second sampling interval. The timestamp in the data indicate the start of the sampling window. We will expand this explanation in the manuscript to make it clear.

**Authors' changes:** We expanded the description (page 8, line 7 onwards).

**RC2, Comment 14:** "P7 L14: I assume „synchronising time stamps" means that the raw time stamps from the DC are rounded to (the nearest?) 10-second time stamps. Update: Okay, further down you write that this is done at SHMI. But what is meant with „synchronising time stamps" then?"

**Authors' response:** See previous comment and response. Essentially, TX and RX time stamps for a given sub-link differs slightly since nodes are not sampled exactly at the same time. The DM synchronizes these time stamps so that they get the same time stamp (i.e. the start of the sampling window). The intention was to always start the sampling window at second 00, 10, 20, 30, 40, 50. However, in reality the sampling window sometimes started a bit early (second 59 etc.). SMHI therefore moved the affected time steps 1 second ahead to obtain a regular time series. We will clarify this in the manuscript.

**Authors' changes:** We clarified this in the manuscript (page 9, line 5).

**RC2, Comment 15:** "P13 L15: I would not say that -30 dBZ „should be considered as zero precipitation". For a standard Marshall-Palmer Z-R relation one would get R=0.04 mm/h for 0 dBZ, which I would considered to be the lower end of what we perceive as rainfall (also in the sense that the hydrometeors are falling and are not suspended in the air, which does not make a difference for the radar). One could debate if 5 dBZ or -5 dBZ is a better threshold, but for sure -30 dBZ is too low."

**Authors' response:** We agree and will modify our wording to better reflect this.

**Authors' changes:** As the referee already suggested an exact threshold is slightly arbitrary, so we went with the suggestion and changed the text to reflect a 0 dBZ as lower threshold for precipitation (page 15, line 16-18).

**RC2, Comment 16:** "P13 Table 5: I had a quick look at the CML data. What I found is that all sublinks with a minimum RX value below -90 dBm had only garbage data. These sublinks have the IDs 208, 389, 391 and 603. The CMLs with min RX values between -85 dBm and -90 dBm (more than 100 CMLs ) seem to look good (note that I only looked at a handful of time series). I am not sure how relevant this is and if this should be stated in the paper, since the dataset is what it is and it is also up to users to investigate all peculiarities. But you might want to consider to add more info on that issue."

**Authors' response:** Your observation is correct for the data below -90dBm. Most probably there is no signal left below -90 dBm, so what is recorded is only noise. For all these time steps, also the corresponding TX (TSL) value is missing. We will add a short explanation in the manuscript.

**Authors' changes:** We added a short explanation in the results (page 15, line 10) and added a sentence in the discussion (page 24, line 4).

**RC2, Comment 17:** "P14 Fig 7: Would it be an option to use a log scale on the y-axis to better show the part of the RX data where rainfall occurs?"

**Authors' response:** Yes, we will plot with a log-scale to highlight the part where rainfall occurs in a better way.

**Authors' changes:** Modified Figure 7

**RC2, Comment 18:** "P19 L3: Looking at the radar images in Fig 13, I do not think that attenuation can cause the missing or too low radar rain rate at the gauge location since there is enough pixels with high rain rates around the gauge pixel. Since there is only one pixel offset between high radar rain rate and the gauge at 15:05, it looks a lot like spatial mismatch due to advection. But, if not yet done, the authors should check the grid definition of the radar grid, i.e. to which corner of the gird pixel (or the centroid) the coordinates refer to. When plotting a quadmesh (like non-equidistant lon-lat grids) the individual pixels are typically defined by the lower left corner."

**Authors' response:** Attenuation is indeed not a very likely cause and at best a very minor contributor, so we will remove this as a possible cause. We will also double check the grid again to make certain no mistakes were made and additionally look at maximum value of the directly neighbouring grid points to see how the precipitation behaviour looks like.

**Authors' changes:** We modified the sentence (page 21, line 3) and double-checked the grid definition for any potential mistakes (we found none).

**RC2, Comment 19:** "P19 L6: In addition to geographic position, also the spatial integration characteristics of the CML might lead to longer lasting but less peaked rain/attenuation events in the time series."

**Authors' response:** Good point, we will modify the manuscript accordingly.

**Authors' changes:** We modified the sentence to add this factor as well (page 21, line 7).

**RC2, Comment 20:** "P19 L9: Since the radar data is not capturing the dynamics of the event correctly (assuming that the gauge and CML do) I would not say that the radar time series „is more realistic" than that of the gauge."

**Authors' response:** We were thinking about the smoothness, but agree that overall radar does not represent the event correctly in this case. We will modify the sentence.

**Authors' changes:** We modified the sentence (page 21, line 10).

**RC2, Comment 21:** "P4 L1: Remove „with""

**Authors' response:** Yes thank you we will remove it.

**Authors' changes:** Removed the error (page 4, line 1).

**RC2, Comment 22:** "P22 L26: Remove „the" before „are provided""

**Authors' response:** Yes thank you we will remove it.

**Authors' changes:** Removed the error (page 24, line 28).

**Reply to Referee #3 (RC3, [https://doi.org/10.5194/essd-2022-221-RC3](https://doi.org/10.5194/essd-2022-221-RC3))**

Dear Referee #3,

Thank you for your review of our manuscript and data set, your recognition of its quality, appreciation of our analysis, and your helpful comments and corrections. Please find below our responses.

**RC3, Comment 1:** "I noticed some +/-1 dB fluctuations in tx with no apparent correlation with rx (just looked at CML 1). I guess it is quantization. Do you have an idea of the reason why quantization is effective only at times?"

**Authors' response:** Yes, these fluctuations are due the resolution that TX power is encoded in (i.e. 1 dB steps). TX fluctuates slightly, for example due to variations in temperature (caused by e.g. variations in radio load and air temperature), which are only sometimes captured at the ±1 dB encoding resolution. Imagine that TX is 10.49 dBm, then the encoded value would be 10 dBm. If TX then increases 0.02 dB (e.g. due to temperature variations), TX goes to 10.51 dBm, and the encoded value ends up to be 11 dBm. On the contrary, if TX is 10.47 dBm, a similar 0.02 dB increase would still result in an encoded value of 10 dBm.

**Authors' changes:** We added an explanation of this (page 13, line 14).

**RC3, Comment 2:** "Sensor altitude data are missing. Can we consider the altitude factor irrelevant everywhere in the area? For instance, link paths are horizontal?"

**Authors' response:** For the CMLs, no authoritative measured data exists on antenna height (only some rough estimates), hence we cannot provide this. However, altitude does not vary a lot in the Gothenburg area, so link paths could be considered nearly horizontal. For the gauges, we provide the altitude of the SMHI station, the altitude of the City gauges is unknown, but often sensors are placed approximately 1.5 m above the ground level. For radar, altitude varies across the domain, depending on measurement angle. Table 3 provides the approximate scan height above the GbgA station. We will clarify in the manuscript.

**Authors' changes:** We clarified this in the manuscript for CMLs (page 6, line 8), and for the City gauges (page 10, line 7).

**RC3, Comment 3:** "About CML data: 10-s resolution is not the CML standard (15-min is more common). It would be really interesting to know whether Ericsson data acquisition system was modified on purpose to carry out rainfall measurements or if this is their standard for network monitoring. Was the firmware of each sensor upgraded to get 10-s data? In general, it's important to know if high temporal resolution set-ups are expensive or not in terms of software/firmware upgrades."

**Authors' response:** Ericsson wrote a software utilizing standard SNMP to carry out the CML data collection at 10 s resolution. No changes were required at the nodes (i.e. the firmware supported this already). Hence, it is not the Ericsson standard resolution for network monitoring, but also not very expensive in terms of software or firmware upgrades. We will clarify this in the manuscript.

**Authors' changes:** We clarified this in the manuscript (page 8, line 7).

**RC3, Comment 4:** "You provided most of this information However, it would be good to have a comparative table of CML, RG, and radar with info about georeferencing (i.e. coordinate system), time axis (origin and format and synchronization if any), temporal resolution and maybe density (sensors/km2)."

**Authors' response:** We do not think such a table would add much, since the information is already provided in either the manuscript or the metadata. Hence, we prefer not to add this.

**Authors' changes:** None

**RC3, Comment 5:** "3, line 32: section should be in Caps"

**Authors' response:** Yes, thank you we will correct the manuscript.

**Authors' changes:** Corrected the word (page 3, line 32).

**RC3, Comment 6:** "6, line 1 (several times since): there should be a space between 100 and m and between 15 and km"

**Authors' response:** Yes thank you, we will correct the manuscript.

**Authors' changes:** Corrected (page 6, line 3).

**RC3, Comment 7:** "10, line 4: included instead of include"

**Authors' response:** Yes thank you, we will correct the manuscript.

**Authors' changes:** Corrected (page 11, line 15).

**RC3, Comment 8:** "10, line 11: "for each" instead of "for each the""

**Authors' response:** Yes thank you, we will correct the manuscript.

**Authors' changes:** Corrected (page 12, line 7).

**RC3, Comment 9:** "10, line 13-14: is this range the min-max difference of Ptx over the observation period?"

**Authors' response:** Yes, we will clarify the manuscript.

**Authors' changes:** We explained this in the sentence (page 12, line 9).

---

## Author Response (AR2)

**Reply to Referee #2 report 19 Oct 2022**

Dear Referee #2,

Thank you for your appreciation of our revised manuscript and data. Please find below our response to your two remaining comments.

**RC2, Comment 1**: "Fig 3.: Since the term „CML" does not appear in the figure it is not 100% clear if one „CML" is one „link". If the term „CML" is not added to the figure, it should be made clear in the figure caption which of the shown parts comprise one CML."

**Authors' response:** We clarify this in the revised figure caption.

**Authors' changes:** Revised caption of Fig. 3.

**RC2, Comment 2**: "L18 and L19: My suggestion would be to use „k" for specific attenuation (dB/km) as it is commonly done in radar meteorology. Since „A" has also been used as symbol for specific attenuation in the CML literature before, this is just a minor suggestion."

**Authors' response:** The literature is indeed not consistent in the use of terms, and sometimes uses the same term for different concepts. However, we have changed to $k$ in Eq. 1 as you suggested and also changed the parameters to make them unique relative to the Z-R relationship (Eq. 4).

**Authors' changes:** Eq. 1 changed.

In addition, we also made some very minor changes to the text and figures to clarify or adapt to the fact that figures had to be submitted as single composites (e.g. Figure 2, Figure 4, section 2.1 and Table 4).